# Characterization of a Family of Scorpion Toxins Modulating Ca^2+^-Activated Cl^−^ Current in Vascular Myocytes

**DOI:** 10.3390/toxins14110780

**Published:** 2022-11-10

**Authors:** Jean-Luc Morel, Nathalie Mokrzycki, Guy Lippens, Hervé Drobecq, Pierre Sautière, Michel Hugues

**Affiliations:** 1Univ. Bordeaux, CNRS, INCIA, UMR 5287, EPHE, F33000 Bordeaux, France; 2Univ. Lille, Institut Pasteur de Lille, F59000 Lille, France; 3Toulouse Biotechnology Institute (TBI), Université de Toulouse, CNRS, INRAE, INSA, 31077 Toulouse, France; 4Univ. Lille, CNRS, UMR 9017, INSERM U1019, CHRU Lille, Institut Pasteur de Lille, Center for Infection and Immunity of Lille, F59000 Lille, France; 5Independent Researcher, F59000 Lille, France; 6Independent Researcher, F33000 Bordeaux, France

**Keywords:** calcium-activated chloride channel, scorpion venom, vascular smooth muscle, animal toxins

## Abstract

The pharmacology of calcium-activated chloride current is not well developed. Peptides from scorpion venom present potent pharmacological actions on ionic conductance used to characterize the function of channels but can also be helpful to develop organic pharmacological tools. Using electrophysiological recording coupled with calcium measurement, we tested the potent effect of peptides extracted from *Leuirus quinquestratus quinquestratus* venom on the calcium-activated chloride current expressed in smooth muscle cells freshly dissociated from rat portal veins. We identified one peptide which selectively inhibited the chloride conductance without effects on either calcium signaling or calcium and potassium currents expressed in this cell type. The synthetic peptide had the same affinity, but the chemical modification of the amino acid sequence altered the efficiency to inhibit the calcium-activated chloride conductance.

## 1. Introduction

Chloride currents are supported by many channels. These are distributed among three different families activated by the membrane potential, the modulation of the volume of the cell [1] and the intracellular calcium concentration [2]. These channels are mainly encoded by the ClC family, a group of structurally-related membrane proteins with 10–12 transmembrane domains [3]. As reviewed recently [4,5], many chloride channels families are expressed in vascular smooth muscle cells. Firstly, volume-regulated Cl^−^ channels (VRCC), activated by hypotonic solution in isolated myocytes from canine pulmonary and renal arteries [6,7], are able to depolarize the membrane of myocytes in response to blood pressure variation [8]. Secondly, Ca^2+^-activated Cl^−^ channels [9,10] that are regulated by inositol 3,4,5,6-tetrakisphosphate (InsP_4_) acting as a second messenger [11,12]. InsP_4_ could represent a negative feedback messenger produced during agonist signalling [13]. The calcium-activated chloride channel is encoded by TMEM16A/Ano1 [14,15,16,17] and is involved in smooth muscle physiopathology in vessels [9,18,19] or myometrium [20,21].

Pharmacologically, anionic channels were targeted by a variety of molecules, such as fenamates, tannic acid, T16A(inh)-A01, CaCC(inh)-A01 [2,22] or cepharantine, after the screening of 530 natural compounds [23].

One class of highly specific and potent inhibitors of the well-characterized families of Na^+^ and K^+^ channels is formed by small proteins isolated from the venom of scorpions belonging to the *Buthidae* family. The basic polypeptides are distributed in two main sized groups. A first group is constituted by long-chain toxins (60–70 residues) with four disulfide bridges, binding specifically to Na^+^ channels [24,25]. The second group consists of short-chain peptides (30–40 residues) with three disulfide bridges, acting on different types of K^+^ channels [26,27]. Four disulfide bridged peptides, such as the short insectotoxins, are also found in the venom of *Leiurus quinquestratus* scorpion species. In this insectotoxin family, chlorotoxin purified from *Leiurus quinquestratus quinquestratus* venom is first described to inhibit enterocyte Cl^−^ channels with an interaction on the cytoplasmic channel face [28] and can interact with the extracellular face of a Cl^−^ channel, expressed in human glioma cells [29,30]. After the chemical synthesis of the polypeptides, we performed a detailed structural analysis of the most potent as a first step towards the functional understanding of these molecules.

## 2. Results

### 2.1. Purification and Characterization of Lqh Peptides

The fractionation by the preparative reverse phase HPLC of the ion-exchange fractions 2, 7 and 8 of the venom of the scorpion *Leuirus quinquestratus hebraeus* was performed according to the procedure described by Marshall et al. [31] and yielded three low molecular weight peptides, named Lqh 2-2, Lqh 7-1 and Lqh 8-6 (Figure 1). 

Capillary electrophoresis for purity assessment followed by mass spectrometry and amino acid analysis revealed that the three peptides have masses of 3676 Da, 3646 Da and 4165 Da, respectively, and are characterized by high contents of cysteine and glycine and low amounts of bulky hydrophobic amino acids; the peptide Lqh 8-6 mainly differs from peptides Lqh 2-2 and Lqh 7-1 by larger amounts of tyrosine and lysine. The full sequence determination (Figure 2) confirmed these data and showed that peptide Lqh 7-1 only differs from peptide Lqh 2-2 by two non-conservative changes at positions 11 (Thr → Met) and 17 (Lys → Glu). Both peptides are closely related to peptide P2 Amm from the scorpion *Androctonus mauretanicus mauretanicus* (77% similarity, [24] and have a lower similarity (around 60%) with peptide Lqh 8-6. The latter appears more closely related to either peptide I from scorpion *Buthus sindicus* with 79% similarity [32] or chlorotoxin from the scorpion *Leuirus quinquestratus quinquestratus* with 71% similarity [28] and insectotoxin I 5A from the scorpion *Buthus cupeus* with 66% similarity [33]. All of these peptides are representative of a peculiar family of short-chain toxins with eight half-cysteine residues distributed in the following consensus sequence: X (0,1) C X(2) C X(10) C X(2) CC X(5,7) C X(4) C X(1) C X(0,3).

### 2.2. Expression of TMEM16A/Ano1 in Rat Portal Vein Smooth Muscle

The ANO1/TMEM16A protein encoding calcium-activated chloride conductance was revealed in rat portal vein smooth muscle by immunolabelling coupled with fluorescence (Figure 3a). Both circular and longitudinal layers expressed ANO1/TMEM16A in the portal vein ring (Figure 3(a1,a2), respectively). 

Moreover, we explored the putative expression of other chloride conductances by RTqPCR to examine whether other chloride conductances were expressed. Based on the expression of reference genes *Ywahz*, *Sdha* and *Hprt*, the level of expression of *Ano1* channel is the highest (2^e(−ΔCt)^ = 0.49) followed by *Gadd45a* and *Slc12a2* (0.154 and 0.126, respectively, Table 1). The RTqPCR suggested a potentially very low expression (0.01 < 2^e(−ΔCt)^ < 0.2, in bold in Table 1) of *Ano6*, *Ano10*, *Clcn6*, *Slc12a4*, *Slc12a6-7*, *Lrrc8a-d*, *Ttyh1*, *Ttyh3* and *Sema3*g and the probable absence of *Ano2-4*, *Ano7-9*, *Clcn1-5*, *Clcn7*, *Clcnka*, *Clcnkb*, *Slc12a9*, *Slc12a1*, *Slc12a5*, *Lrrc8e*, *Ttyh2* and *Tmem206* (Table 1: 2^e(−ΔCt)^ ≤ 0.01). 

### 2.3. Inhibitory Effects on Calcium Activated Chloride Current Expressed in VSMC

In rat portal vein myocytes, at a holding potential of −50 mV, the external application of 10 mM caffeine induces a transient current depending on an increase in cytoplasmic Ca^2+^ concentration, this current was characterized as Cl^−^ current [9] in the used experimental conditions. The amplitude of the Ca^2+^-activated Cl^−^ current in our setup was 190 ± 20 pA (*n* = 101). The external application of Lqh 7-1 (1 µM) over 10 min inhibited the Ca^2+^-activated Cl^−^ current without affecting the caffeine-induced Ca^2+^ response (Figure 3b; maximal amplitude of caffeine-induced Ca^2+^ responses in control condition 281 ± 32 nM versus 278 ± 28 nM, *n* = 32, *p* = 0.33). The IC_50_ value for the inhibition of Lqh 7-1 was determined to be 63 ± 13 nM (Figure 3c). Synthetic Lqh 7-1 peptide (see material and methods) was found to be as potent as the native Lqh 7-1 with an IC_50_ value of 49 ± 5 nM (Figure 3c). Despite its high degree of sequence identity, Lqh 2-2 only inhibited 50% of the current at a 1 µM concentration while Lqh 8-6 had no effect on the Ca^2+^-activated Cl^−^ current, despite its high sequence homology with Lqh 7-1 (Figure 3c). The external application of 1 µM of chlorotoxin inhibited only 36 ± 9% of the Ca^2+^-activated Cl^−^ current. 

### 2.4. Specificity of Lqh 7.1 Peptide

We verified Lqh 7-1 specificity on different currents supported by channels expressed on myocytes. The Lqh7-1 peptide inhibited the chloride conductance (Figure 4a, *n* = 16 cells from 3 different cell dissociations were tested for each concentration, the maximal amplitude of the current was significantly decreased from 230 ± 40 pA to 110 ± 21 pA and 59 ± 11 pA in presence of 0.1 and 0.5 µM of Lqh7-1 toxin, respectively) without the alteration of the amplitude and frequency of the Ca^2+^-activated non-selective cationic current as described on portal vein myocytes as measured after during and after the chloride current deactivation [35]. 

Short applications of 10 µM Norepinephrine induced a transient increase in cytosolic Ca^2+^ concentration due to the mobilization of intracellular store by inositol 1,4,5-trisphosphate) and were not affected by the presence of 0.5 µM of Lqh 7-1 (Figure 4b, tested before and after toxin application with a delay of 10 min between both NE applications, maximal amplitude 232 ± 8 nM in control conditions versus 224 ± 10 nM in the presence of toxin, *n* = 36 cells from 3 different dissociations, *p* = 0.42). Similar results were obtained with application of 10 nM angiotensin II, inducing a transient increase in cytosolic Ca^2+^ concentration via the activation of the L-type Ca^2+^ channel and the Ca^2+^-induced Ca^2+^ release due to ryanodine receptors [36]. The maximal amplitude 275 ± 21 nM in control conditions versus 263 ± 31 nM in the presence of toxin, and *n* = 40 cells from three different dissociations, *p* = 0.4. Finally, the acetylcholine-induced Ca^2+^ response was not modified in presence of 0.5 µM of the Lqh7-1 toxin (maximal amplitude 232 ± 11 nM in control conditions versus 221 ± 8 nM in the presence of toxin, and *n* = 30 cells from three different dissociations, *p* = 0.43).

In portal vein myocytes, the voltage-activated Ca^2+^ currents integrated the function of L-type and T-type Ca^2+^ channels and can be evaluated though Ba^2+^ currents activated by depolarization; both the amplitude and kinetic parameters of Ba^2+^ currents were not affected by the presence of Lqh 7-1 peptide, as illustrated in Figure 4c. To separate L-type and T-type voltage-dependent Ca^2+^ channels, depolarization from −60 to 0 mV was applied at 0.05 Hz or by a short depolarizing step from −80 to 0 mV was applied at 0.2 Hz, respectively. The bath solution KCl (130 mM) and CaCl_2_ (2 mM) were replaced by CsCl (130 mM) and BaCl_2_ (5 mM), respectively, whereas in patch-clamp pipette, KCl was replaced by CsCl [37,38]. Lqh 7-1 (0.5 µM) did not affect both L-type and T-type voltage-dependent Ca^2+^ channels (maximal amplitude 260 ± 41 pA in control conditions versus 259 ± 47 pA in the presence of Lqh 7-1 peptide, *n* = 11, and maximal amplitude 139 ± 32 pA in control conditions versus 127 ± 38 pA in presence of Lqh 7-1 peptide, *n* = 5, respectively). 

For K^+^ current recording, the patch-clamp pipette contained 130 mM KCl and the external solution contained 5.6 mM KCl. Ca^2+^-activated and voltage-activated K^+^ currents were elicited by depolarizing pulses from −80 to 0 mV and Ca^2+^-activated K^+^ currents were activated by the opening of Ca^2+^ channel during the depolarization step. The K^+^ currents were not affected by 0.5 µM Lqh 7-1 (*n* = 9). The presence of both type of K^+^ currents was confirmed by their sequential inhibition by charybdotoxin (100 nM) and scyllatoxin (100 nM). 

### 2.5. Modeling of Lqh 7-1 Toxin

The primary structure of the polypeptides agrees with the characteristic disulfide bridge-stabilized a/b pattern [39]. Therefore, we thought it of great interest to see what surface elements contributed towards the highly potent and specific interaction of the toxin with the chloride channel. The tertiary structure of the most potent peptide, Lqh 7-1 was modeled by AlphaFold, and contains a short three-stranded β-sheet and an α-helix, linked together by a three disulfide bridge (Figure 5a). The N-terminal strand is linked to the helix by a fourth disulfide bridge. This topology agrees with that observed for chlorotoxin and similar peptides, as indicated in Figure 2.

Comparative electrostatic calculations starting from the 3D structures of chlorotoxin and charybdotoxin put forward the hypothesis that the region of the α/β turn is implicated directly in the interaction with the Cl^−^ channel. The topology of this region is not conserved between chlorotoxin and Lqh 7-1, leading to substantial differences in electrostatic potential (Figure 5a) and possibly explaining their differential activity. 

Moreover, although it is without apparent modification on the structure as detected by circular dichroism, the key mutation between Lqh 7-1 and Lqh 2-2, K_17_ → E changes the electrostatic potential in the α/β region considerably. This fact, together with the observed ten-fold decrease in activity (Figure 5b,c), leads us to conclude that the interaction between this family of toxins and the Cl^−^ channels indeed involves the residues in the region of the α/β turn. All the mutants were tested on caffeine-induced currents and none of them is able to alter the recorded current after 10 min of incubation (Figure 5).

## 3. Discussion and Perspectives

Chloride channels form a wide heterogenous family of anionic channels [1,2,13,40]. Functionally, anionic channels could be activated by cell volume changes or the changes of intracellular concentrations of proton, InsP4 or Ca^2+^. The Ca^2+^-activated chloride channel named TMEM16A/Ano1 is involved in several pathologies as hypertension, stroke and cancer and was expressed in several cell types in mammalians, including smooth muscle and epithelial cells, as reviewed recently [41,42,43]. Immunostaining and RTqPCR results suggested that this channel could support the Ca^2+^-activated chloride current observed in smooth muscle cells from rat portal veins. Previously, this conductance was characterized and blocked by anthracene-9-carboxylic acid and diphenylamine-2,2′-dicarboxylic acid [9] also reported as inhibitors of TMEM16A channels [22]. Even if the anthracene-9-carboxylic acid was used on TMEM16A channel transfected in HEK-293T cells to define the function of the channel [44], the known antagonists—including anthracene-9-carboxylic acid—were poorly specific yet tested in preclinical trials to become pharmacological drugs [41,45]. Toxins from venoms are used more often as preclinical tools to find pharmacological drugs than therapeutic molecules. In this work, we describe a new family of peptidic toxins which specifically interact with Cl^−^ channels. Lqh 7-1 and Lqh 2-2 inhibit a Ca^2+^-activated Cl^−^ current on rat portal veins with specificity and affinity close to the best organic drugs targeting this channel [22,23]. As previously described, our results confirm that chlorotoxin is not able to inhibit TMEM16A channels (near to 20% of inhibition at 1 µM) [46]. The most potent of these, Lqh 7-1, inhibits the Ca^2+^-activated Cl^−^ current with an IC50 in range of 50 nM and is inefficient on Ca^2+^ and K^+^ conductances at concentrations of 1 µM. Moreover, the toxin did not alter intracellular calcium signaling activated pharmacologically (caffeine application) as well as via the activation of G protein-coupled receptor. Although Lqh 7-1 appears to be more specific for the Ca^2+^-activated Cl^−^ current, our comparative experiments indicate that subtle differences limited to the level of a single amino acid can alter the electrostatic potential and hence the interaction between the channel and the peptide. The Lqh 7-1 toxin is easily synthesized and the synthetic toxin has the same affinity for the Ca^2+^-activated Cl^−^ channel as well as the same inhibition potency on Ca^2+^-activated chloride current than native toxin. To better understand the link between the toxin and its target, it is necessary to radiolabel the peptide. Because the addition of an iodine 125 altered the inhibitory effects of the native or synthetic Lqh 7-1, we performed sequence modifications. Unfortunately, all modifications suppressed the inhibition of the current by the toxin. We demonstrate that specific amino-acid changes by toxin synthesis, which do not alter the three-dimensional structure, could abolish toxin inhibition activity on Ca^2+^-activated Cl^−^channels. These results suggest an important role of these amino acids in its functionality. The toxin can be included in the superfamily of cystine dense peptides that are easy to synthesize and present specific properties, such as thermal, chemical and proteolytic stability, that are required to be usable as pharmacological drugs. Our data increase the number of toxins acting on anionic channels from *Leiurus quinquestriatus hebraeus* venom reported in a STEP database as chlorotoxin or GaTx1 and propose a new family of toxins acting on TMEM16 proteins known as chloride channels. These channels are also regulated by lipids, such as cholesterol and phosphatidylinositol 4,5-bisphosphate [47,48], and some authors have proposed that they are key regulators of phospholipid repartition in the plasma membrane [42]. At this stage of investigation, we are not able to indicate whether this toxin’s family is able to discriminate both functions of TMEM16 proteins. As proposed recently, BsTx7, BsTx14, GaTx and chlorotoxin contain the CCGG motif that is able to interact with MMP2 to inhibit its activity and induce its internalization [34,49]. The consequence of this interaction could participate in the inhibition of the specific chloride current observed in glioma cells, thereby giving anti-metastasis and anti-tumor effects to these toxins. Lqh toxins, described here, also present this CCGG motif, suggesting that they can also be used to counter glioma [50]. Moreover, if the toxin internalization is supposed to be necessary to inhibit the chloride current, it is possible that modifications introduced in the Lqh 7-1 sequence are responsible for the loss of the internalization process and consequently the alteration of Lqh7 efficacy. The peptide family (described in Figure 2) also suggests that the specificity of each toxin to its target is probably due to the subtle modifications in the sequence and consequently in the structure. Because other toxins, such as BsTx7 and chlorotoxin, can interact with other proteins, and some molecular tools were derived from chlorotoxin to visualize and reduce cancer progression [51,52,53,54]. The inhibitory effects of Lqh7 on glioma cell proliferation should be evaluated.

In the future, because TMEM16A participates in the myogenic tone of cerebral artery [55] and in pericyte contraction [56], it could become a target in small vessel disease, stroke or vascular dementia and also in systemic or pulmonary hypertension by controlling vasoconstriction [57,58] or the vascular remodeling [18,19,59]. Furthermore, TMEM16A is expressed in endothelial cells and its expression can ameliorate hypertensive phenotypes induced by angiotensin-II infusion [60] or angiogenesis [61]. The toxins represent very important new drug design tools because the function of TMEM16A in vascular smooth muscle cells is highly dependent on the vascular bed, as suggested in TMEM16A-KO mice presenting a decrease in the aorta and increase of saphenous arteries [17]. The inhibition of TMEM16A is also useful for reducing the progression of renal fibrosis in polycystic kidney disease by decreasing intracellular chloride [62]. TMEM16A could also target brain functions; as recently reported the specific KO of TMEM16A in the habenula which induced the decrease of cholinergic neuron activity to alter behaviors, as observed in autistic spectrum disorders [63].

## 4. Materials and Methods

### 4.1. Purification of Lqh’s Peptides

The fraction 7 obtained from the ion-exchange chromatography of venom from the scorpion *Leuirus quinquestratus hebraeus* (crude venom was from Latoxan) was fractionated by reverse-phase HPLC on a Nucleosil C18 column (5 µm, 100 A pore size, 500 × 10 mm) and equilibrated with 0.1% trifluoroacetic acid. Elution monitored at 220 nm and 280 nm was performed with a stepwise gradient of acetonitrile in 0.1% trifluoroacetic acid: 0–30% for 15 min and 30–50% for 40 min at a flow rate of 2.5 mL/min. Peptides Lqh 2-2, Lqh 7-1 and Lqh 8-6 were eluted with retention times of 38, 23, 35 min, respectively, under these conditions. The purity of the polypeptides obtained was assessed by capillary electrophoresis (270 A-HT Capillary Electrophoresis System, Applied Biosystems) in 20 mM sodium citrate buffer pH 2.5 at 30 kV and 30 °C for 10 min using a 50 cm capillary, 50 µm in diameter. Samples were injected for 1 sec in a 5-inch vacuum. UV detection was performed at 220 nm. The procedures for purification were similar to those classically used [38,64].

### 4.2. Ion Spray Mass Spectroscopy

The molecular masses of the native polypeptides were determined by plasma desorption and ion-spray mass spectrometry. Plasma desorption mass spectra were recorded on a Bio-Ion 20 ^252^Cf fission fragment ionization lime-of-flight mass spectrometer (Thermofisher, Applied Biosystems, Waltham, MA, USA). Approximatively 10–20 µL of each fraction of HPLC were applied onto nitrocellulose-backed foil. The spectra were accumulated during 10^6^ fission events (about 15–20 min). For ion spray mass spectrometry, freeze-dried samples of native polypeptides were dissolved in 20% acetonitrile and 0.1% formic acid in water at a concentration of 20 pmol/µL. Ion spray mass spectra were recorded on a simple-quadrupode mass spectrometer API I (Perkin-Elmer, Waltham, MA, USA) equipped with an ion spray (nebulizer-assisted electrospray) source (Sciex, Toronto, ON, Canada). The solutions were continuously infused with a medical infusion pump (Model 1 1, Harvard Apparatus, South Natick, MA, USA) at a flow rate of 5 µL/min. Polypropylene glycol (PPG) was used to calibrate the quadrupole. Ion spray mass spectra were acquired at unit resolution by scanning from *m*/*z* 1200 to 2400 with a stop size of 0.1 Da and a dwell time of 2 ms. Five to ten spectra were summed. The potential of the spray needle was held at +5 kV. Spectra were recorded at an orifice voltage of +90 V. Mac BIO Spec was the computer program used for the calculation of the molecular masses of the samples. 

### 4.3. Determination of Lqh’s Sequences

Peptides sequences were determined using an online phenylthiohydantoin amino acid analyzer (Applied Biosystems 120A). Polypeptides were reduced for 4 h at 37 °C under argon in 0.1 M Tris/HCl buffer at pH 8.3, containing 6 M guanidinium chloride and 0.1 M 2-mercaptoethanol, and then alkylated at 37 °C for 2 h under argon in the dark by adding iodoacetamide to 0.2 M (final concentration). The carboxamidomethylated polypeptides were then desalted by RP-HPLC on a C8 column (7 µm, 300 A pore size, 30 × 4.6 mm) using a linear gradient of acetonitrile from 0 to 60% in 0.1% trifluoroacetic acid for 30 min at a flow rate of 1 mL/min. The carboxamidomethylated polypeptides (2 nmol) were digested in either 0.1 M ammonium bicarbonate at pH 8.0 at 37 °C for 4 h with carboxypeptidase B using an enzyme-to-substrate ratio 1:10 (by mass) for Lqh 2-2 and Lqh 7-1 or in 50 mM sodium acetate at pH 4.0 at 30 °C for 5 and 10 min with carboxypeptidase B using an enzyme-to-substrate ratio of 1:100 (by mass) for Lqh 8-6. The released amino acids were analyzed using the amino acid analyzer.

### 4.4. Synthesis of Lqh 7-1

The Lqh 7-1 toxin was synthetized using solid-phase t-Boc-Benzyl strategy on MBHA resin (0.5 mmols). The synthesis was performed on a synthesizer ABI430 using home-made in situ neutralization cycles. Activation was performed with HBTU. After HF cleavage, the reduced toxin was purified by reverse phase HPLC onto C18 Nucleoprep column (500 × 10 mm; Shandon). Homogeneous fractions in mass spectrometry, analytical HPLC and cappilary electrophoresis were pooled and refolded in Tris 0.1 M buffer at pH 8.5 containing Cys-Cys/Cys 4 mM/2 mM at 4 °C for 20 h. The medium was then acidified to pH 4.0 with TFA, filtered through Millipore HAWP 0.45 m filter and loaded onto a C18 Nucleoprep column. The separation was performed with a linear gradient from 10 to 60% of acetonitrile at 60 °C to overcome the problem of the slow isomerization of proline3. Pure fractions co-cluting with the native toxin were pooled and lyophilized.

### 4.5. Circular Dichroism

Circular dichroism was performed with a Jobin-Yvon CD6 dichrograph in 0.2–1 cm pathlength devices after the adjustment of peptide concentrations and performed as previously described [65,66]. For the determination of the helicity, we resorted to the method described [67] and based on the measured circular dichroism at 220 nm. 

### 4.6. Preparation of RNA and RT-PCR

After the disruption and homogenization of all samples with minilys (Precellys, distributed by Ozyme France, Montigny le Bretonneux) in a tri-reagent (Molecular Research Center, Inc., Cincinnati, OH, USA), the isolation of the total RNA from both smooth muscle layers from four rat portal veins was performed following the supplier procedures. The concentration of RNA was measured with spectrophotometry (NanoDrop Technologies, Wilmington, DE, USA) for each sample. Subsequently, cDNA was sythetized with Iscript gDNA clear DNA systhesis kit (Bio-Rad, 172-5035, Hercules, CA, USA) and the qPCR was performed using SsoAdvanced Universal Sybr Green Supermix (Bio-Rad, 172-5271), following the protocols of the supplier. The qPCR was carried out using a 96 well plate with 48 unique assays containing primers to detect rat chloride channels indicated in Table 1 (Bio-Rad custom plate 10025219) as recommended by the supplier and stopped after 40 cycles. The list of the primers amplicons is indicated in the Appendix A.

### 4.7. Immnohistochemistry Coupled to Fluorescence

The animals did not undergo any procedures that were deleterious to their well-being before being euthanized. They were adult male (8- to 12-week-old) supernumerary animals related to the production of the animal facility. The animals were euthanized by CO_2_ inhalation in a dedicated chamber and the portal veins were dissected. The tissues were fixed directly in PB buffer solution (4.8 g/L sodium phosphate monobasic Merck, S0751 and 22.72 g/L and sodium phosphate dibasic Merck, Saint Louis, MO, USA, S0876) containing paraformaldehyde (40 g/L; Merck, P6148). After 30 min, the portal veins were placed in PB solution containing sucrose (200 g/L; Euromedex 200-301-B) for 6 h and placed in OCT (thermofisher scientific LAMB/OCT) and frozen. The portal vein was cut perpendicularly to the vessel axis with a cryostat (Leica CM3050S) and the 50 µm thick rings were placed in a PB buffer containing sodium azide (2 g/L; Merck, S8032) before immunostaining. Vessel rings were rinced twice in PBS for 10 min on a shaker and placed in blocking solution (PB buffer containing 2% goat serum, Eurobio CAECHV00-0U, 0.2% Triton x-100, Merck X-100 and 1 g/L bovine serum albumin, Euromedex 04-100-812-C) for 1 h. The primary antibody directed against TMEM16A/Ano1 (GeneTex, GTX64457) was diluted in blocking solution 1:100 and the portal vein rings were incubated in it overnight (14 h). Tissues were rinsed 3 times in blocking solution and incubating for 2 h in blocking solution containing the secondary antibody (anti-anti-rabbit IgG Dylight488 conjugated, dilution 1:500, Bethyl A120-108-D2). The samples were rinsed 3 times in PB buffer, incubated in DAPI (1:50,000; Interchim, FP371867), rinsed again in PB buffer and mounted in fluoromount-G (Interchim, FP-483331). After 72 h, imaging was performed with a DMI6000-SP5 confocal microscope (Leica microsystems).

### 4.8. Animals

Animals to isolate the cells, tissues were taken from rats (150 to 250 g of body weight) which were euthanized in a CO2 chamber according to the protocol defined by the manufacturer (CO2 rack, Tem-Sega, Pessac, France) in a dedicated room of the animal facility (A32-063-940). in accordance with the European directive (n°2010/63/UE, revising n°86/609/CE). All experiments used rats from a supplier (Janvier labs, Le Genest-Saint-Isle, France) and supernumerary rats produced in the animal facility. Tissues were collected in HBBS medium at 4 °C and transported to the laboratory for tissue dissection, cell preparation and RTqPCR experiments. 

### 4.9. Cell Preparation

Isolated myocytes from rat portal veins were obtained by enzymatic dispersion, as described previously (Morel et al. 1996). Cells were maintained in short-term primary culture in M199 medium containing 2% fetal calf serum, 2 mM glutamine, 1 mM pyruvate, 200 units/mL penicillin and 200 µg/mL streptomycin; they were kept in an incubator gassed with 95% air and 5% CO_2_ at 37 °C and used within 24 h.

### 4.10. [Ca^2+^]_i_ Measurements

Cells were loaded by incubation into a physiological solution containing 1 µM Indo-1-acetoxymethylester for 30 min at room temperature. These cells were washed and allowed to cleave the dye to the active Indo-1 for 30 min. Indo-1-loading was usually uniform over the cytoplasm, and the compartmentalization of the dyes were never observed. The measurement of [Ca^2+^]_i_ in single cells and calibration curves for Indo-1 were determined within cells, as previously reported (Morel et al. 1996). Indo-1-loaded cells were mounted in a perfusion chamber and placed on the stage of an inverted microscope (Nikon Diaphot, Tokyo, Japan). All measurements were taken at 25 ± 1 °C.

### 4.11. Membrane Current and [Ca^2+^]_i_ Measurements

Voltage-clamp and membrane current recordings were made with a standard patch-clamp technique using a List EPC-7 patch-clamp amplifier (Darmstadt-Eberstadt, Germany). Patch pipettes had resistances of 3–4 MΩ. When [Ca^2+^]_i_ measurements were carried out simultaneously, Indo-1 (50 µM) was added to the pipette solution in the whole-cell recording mode and [Ca^2+^]_i_ was estimated from the 405/480 nm fluorescence ratio, using a calibration determined within cells, as previously described [36]. Patch-clamp experiments were carried out at 30 ± 1 °C.

### 4.12. Solutions

The normal physiological solution contained (in mM): NaCl 130, KCl 5.6, MgCl_2_ 1, CaCl_2_ 2, glucose 11 and HEPES 10, pH 7.4 with NaOH. The basic pipette solution contained (in mM) CsCl 130, HEPES 10 and pH 7.3 with NaOH (CsCl was replaced by KCl for potassium current measurements). Caffein, and other active compounds, were applied to the recorded cell by pressure ejection for the period indicated on the records. Lqh peptides, chlorotoxin and other toxins were added in a bath solution. 

### 4.13. Chemicals and Drugs

Collagenase was obtained from Worthington (Freehold, NJ, USA); pronase (type E), bovine serum albumin and norepinephrine were from Sigma (St Louis, MO, USA). M199 medium was from Flow Laboratories (Puteaux, France). Fetal calf serum was from Flobio (Courbevoie, France). Streptomycin, penicillin, glutamine and pyruvate were from Gibco (Paisley, UK). Caffeine was from Merck (Nogent sur Marne, France). Indo-1 and Indo-1AM were from Calbiochem (Meudon, France). Angiotensin II was from Neosystem Laboratories (Strasbourg, France). Chlorotoxin was from our laboratory (Service des biomolécules, Institut Pasteur de Lille, Lille, France). Charybdotoxin and scyllatoxin were from Latoxan (Rosans, France).

### 4.14. Data Analysis

The pooled data of n experiments are given as mean ± standard error. Significance was tested with Student’s t-test. The level of significance is 0.05. Inhibition-response curves were analyzed by a nonlinear least-squares fitting program according to models involving one or two binding sites (Graphpad-Prism software).

## Figures and Tables

**Figure 1 toxins-14-00780-f001:**
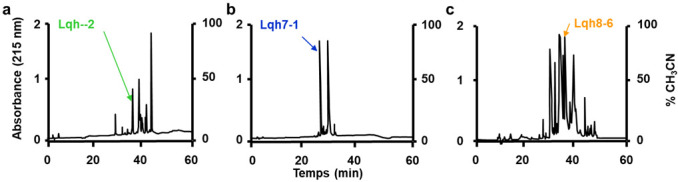
HPLC chromatogram showing (**a**) Lqh 2-2, (**b**) Lqh 7-1 and (**c**) Lqh 8-6 peptides selected to be tested on vascular smooth muscle cells from rat portal veins.

**Figure 2 toxins-14-00780-f002:**
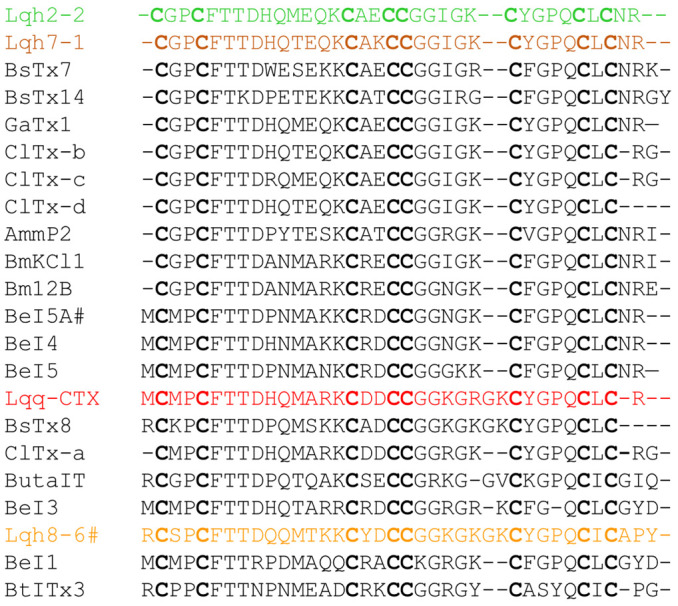
Full sequence of Lqh peptides and comparison with other closed known sequences described in [34]. The color code was applicated on all figures. Lqh toxins and GaTx1, known as CFTR inhibitors, were from *Leuirus quinquestratus hebraeus,* Lqq-CTX (Chlorotoxin) from *Leuirus quinquestratus quinquestratus*; AmmP2 from *Androctonus mauretanicus mauretanicus*; BsTx from *Buthus sindicus;* BeI from *Buthus cupeus;* ButaIT from *Mesobuthus tamulus*; BtITx3 from *Hottentotta tamulus*; BmKCl1 and Bm12B from *Mesobuthus martensii*; and ClTx from *Odontobuthus doriae*.

**Figure 3 toxins-14-00780-f003:**
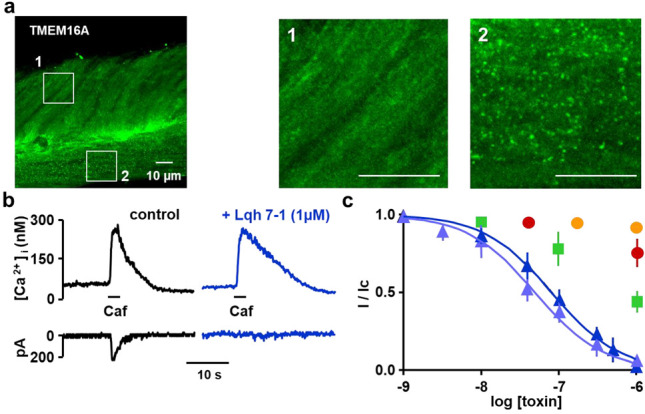
(**a**) typical immunolabelling with anti-TMEM16A revealed by confocal microscopy in rat portal vein slice. The experiments were reproduced on dissociations from three different rats. (**b**) Effect of Lqh 7-1 (1 µM) on typical Ca^2+^-activated Cl^−^ current activated by caffeine application in control conditions (black) and in presence of the peptide (blue). (**c**) Effect concentration curves of Lqh 2-2 (green), native Lqh 7-1 (dark blue), synthetic Lqh 7-1 (blue), chlorotoxin (red) and Lqh 8-6 (orange). *n* = 7–31 cells per concentrations reported in the graph (**c**), cells were provided from five different dissociations/rats.

**Figure 4 toxins-14-00780-f004:**
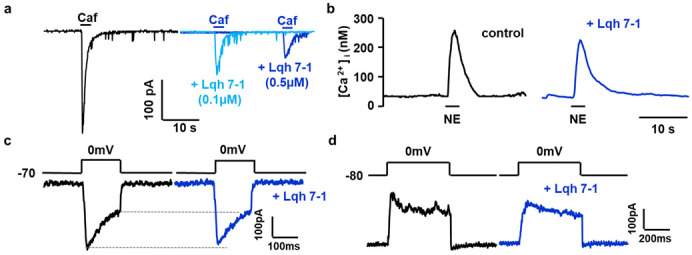
Effect of Lqh 7-1 peptide (0.1 and 0.5 µM) (**a**) on a representative Ca^2+^-activated current in presence of Cesium chloride to inhibit K^+^ currents; two different traces with two different concentration of Lqh 7-1 were superimposed (**b**) on norepinephrine-induced Ca^2+^ responses, (**c**) on voltage activated Ba^2+^ current through voltage gated Ca^2+^ channels as described by Loirand et al. [35] and (**d**) on a voltage-activated K^+^ current activated by depolarization. Dark blue traces illustrated measurements were recorded in presence of peptide Lqh 7-1 (0.5 µM).

**Figure 5 toxins-14-00780-f005:**
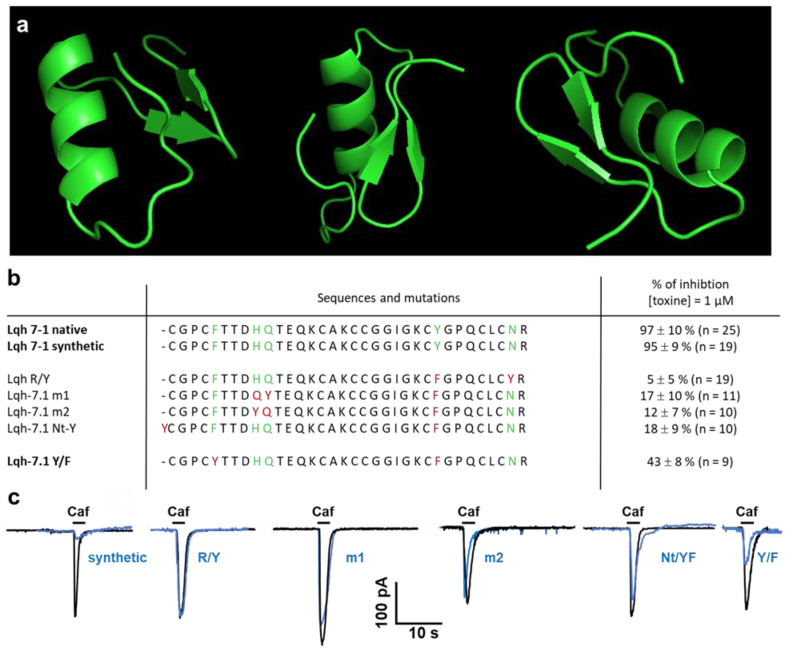
(**a**) Model structure of the native Lqh7-1 obtained by AlphaFold [40], (**b**) sequences and inhibition of Ca^2+-^activated chloride current evoked by caffeine application on vascular smooth muscle cells. (**c**) Representative Ca^2+-^activated currents evoked by caffeine application in control conditions (black) and after 10 min of the application in the perfusion bath of the Lqh-7.1 toxin mutants (in blue, *n* = 9–15 cells were recorded in the absence and presence of 1 µM of toxin mutants). Traces were superimposed and the pipette solution contains cesium chloride to inhibit K+ currents. The lightning bolts indicate the point where the mutations are made. The PDB files of the models are given in the Appendix A.

**Table 1 toxins-14-00780-t001:** Expression of genes encoding proteins encoding chloride conductance using RTqPCR approach with known primers in rats. Values are 2^e(−ΔCt)^ to indicate the level of expression in comparison with reference genes.

*Ano2*	*Ano3*	*Ano4*	*Ano6*	*Ano7*	*Ano8*	*Ano9*	*Ano10*	*Cftr*
0.007	0.0007	0.01	0.08	0.003	0.01	0	0.07	0.003
*Clcn1*	*Clcn2*	*Clcn3*	*Clcn4*	*Clcn5*	*Clcn6*	*Clcn7*	*Clcnka*	*Clcnkb*
0	0.005	0	0	0.0002	0.02	0.002	0.0004	0.001
*Slc12a1*	*Slc12a2*	*Slc12a4*	*Slc12a5*	*Slc12a6*	*Slc12a7*	*Slc12a9*	*Sema3g*	*Sadd45a*
0.001	0.126	0.044	0.0001	0.03	0.04	0	0.057	0.154
*Lrrc8a*	*Lrrc8b*	*Lrrc8c*	*Lrrc8d*	*Lrrc8e*	*Ttyh1*	*Ttyh2*	*Ttyh3*	*Tmem206*
0.065	0.045	0.020	0.028	0	0.077	0	0.019	0.004

## Data Availability

The data presented in this study are available in this article and Appendix A.

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
