# Peer review of "Characterization of a Family of Scorpion Toxins Modulating Ca2+-Activated Cl− Current in Vascular Myocytes"

_toxins, 2022, doi:10.3390/toxins14110780_

Round 1
Reviewer 1 Report
The manuscript purified peptide toxins from scorpion venom and identified Lqh 7-1 which the authors claim to be an inhibitor to Ca2+-activated Cl- current in vascular myocytes. Though the results might be potentially interesting, the current conclusion is not well supported by the experimental data.
Major concerns:
1. The electrophysiological currents shown in Figure 3 and Figure 4 may not be the currents of different ion channels the authors claim. There are no unarguable evidence to suggest that these currents are from calcium-activated Cl channels, voltage gated calcium channel and calcium-activated potassium channels. As there are many subtypes of currents in smooth muscle cells, the authors need to be extremely careful in making any conclusion and should provide solid experimental evidence.
2. There is no any statistical figure in figure 4, it is even harder to judge how reliable these data are.
3. When the authors tried to prove the specificity of Lqh 7-1, they used different concentrations of this peptide against different types of ion channels they believe, which makes the comparison not meaningful. The concentration of the peptide should be consistent in all representative figures for electrophysiological recordings.
4. The representative electrophysiological recordings of each mutant of the peptide used to target the chloride channel (Figure 5) should be shown directly. The current version is too descriptive and preliminary without any mechanistic insight about the interaction between the channels and the peptide.
Author Response
The authors thank the three experts who took the time to provide their expertise to improve the quality of this manuscript. They will find our answers and arguments hereafter in blue and an amended manuscript with the modifications in red.
For all reviewers we want to precise why you have submitted this manuscript. This study began in 1997, the selectivity of toxins from scorpion venom for ion channels, gave us the idea to look for a toxin that could inhibit the activated chloride-calcium conductance of smooth muscle cells. At that time the Ano1 channel had not been identified and it that why we have performed experiment on the cellular model well known in the lab with clearly electrophysiologically identified chloride conductance. This work was presented in several congresses but had not yet been published because we did not find the structural modification of the toxin allowing to derive a radioligand for precisely identifying the target. We have recently reviewed this work and have decided to submit it to your judgment because 1/ the interest of targeting activated chloride-calcium channels of smooth muscle cells is making a comeback (see bibliography of the MS) and 2/ the study by RTqPCR and by immunohistological labeling of the chlorine conductance of the cellular model used suggests that Ano1 is clearly expressed and 3/ our work confirms that the position of this toxin family as a ligand of the channels carrying chlorine conductance or of their direct regulator. Finally, the structural modifications that we have made also suggest a very precise interaction between toxin and target protein showing the subtlety of protein interactions.
Reviewer 1
The manuscript purified peptide toxins from scorpion venom and identified Lqh 7-1 which the authors claim to be an inhibitor to Ca2+-activated Cl- current in vascular myocytes. Though the results might be potentially interesting, the current conclusion is not well supported by the experimental data.
Major concerns:
- The electrophysiological currents shown in Figure 3 and Figure 4 may not be the currents of different ion channels the authors claim. There are no unarguable evidence to suggest that these currents are from calcium-activated Cl channels, voltage gated calcium channel and calcium-activated potassium channels. As there are many subtypes of currents in smooth muscle cells, the authors need to be extremely careful in making any conclusion and should provide solid experimental evidence.
The cellular model was chosen because of the characterization of the nature of the ionic currents which had been carried out in this cell type (Baron et al., 1991; Loirand et al., 1991, 1989). Thus the ionic currents activated by calcium are mainly chloride and potassium currents, the presence of Cs in the intracellular medium causing the inhibition of potassium conductance to assume that the residual currents are chloride one as characterized before (Baron et al., 1991). Moreover, for tested voltage dependent current , we have used electrophysiological condition and intracellular and extracellular media to isolate barium currents supported by voltage dependent calcium channels (Loirand et al., 1989). We have chosen to illustrate the conditions in which the L-type calcium current is principally activated. It is the most important conductance in this cell type as the Mironneau’s group (but not only) had shown. We have tested also the Lqh7 toxins in this cell type with electrophysiological conditions required to isolated T-type barium current. The toxins have no effects and we have not illustrated it. We have also tested the toxin on voltage activated currents expressed in cardiomyocytes and the toxin had no effects on these currents (we have chosen to not illustrate this part of the work to reduce the “no effect” part.
This study was began in 1997, and our purpose was not to test the toxin on all known ion channels to be expressed in vascular smooth muscle cells, but test the effects of the toxin on the most important conductances separable on the VSMC from rat portal vein, the well-known VSMC model in the lab. We propose to modify the legend of the figure 4 to recall the experimental conditions and the references to precise why we suggest that are chloride, potassium and calcium currents.
- There is no any statistical figure in figure 4, it is even harder to judge how reliable these data are.
We add statistics and number of experiences in the result chapter (line 120-127)
When the authors tried to prove the specificity of Lqh 7-1, they used different concentrations of this peptide against different types of ion channels they believe, which makes the comparison not meaningful. The concentration of the peptide should be consistent in all representative figures for electrophysiological recordings.
We have modified figure 4 because indeed it was not very clear that the toxin was used at 0.5µM to look for non-specific effects on other conductances. and we have chosen to add the dose of 0.5 µM on the a/ panel to be more precise. On this panel the decrease of chloride current is visible and the absence of effect on amplitude and frequency of store operated small inward current are not affected. We have also corrected the scale bar for the figure 4.
We also add quantification in the main text. we hope that the modifications will satisfy the reviewer.
- The representative electrophysiological recordings of each mutant of the peptide used to target the chloride channel (Figure 5) should be shown directly. The current version is too descriptive and preliminary without any mechanistic insight about the interaction between the channels and the peptide.
We have added some experimental traces with and without mutant applications. We completely agree with the expert's remark, we are not able to give a mode of action of this toxin and it is to be noted with great regret. The purpose of the mutations of the toxin was to be able to derive a radioligand from it in order to study its interactions with the channel, but the labeling with radioactive iodine could never be carried out because of the loss of the bond by the different possible mutations.
Finally, I would add that the electrophysiological tests were all conducted double-blind. After these multiple attempts at mutations, we took the decision to stop this program. I draw attention to the fact that most of the mechanisms of action of these scorpion toxins modulating chloride conductances are not known, probably because of particular interactions with the channels or proteins associated with them. Our work improves the knowledge on these families of toxins and it seems to me that it will allow the scientific community to imagine new means of studying these chloride conductances whose interest as pharmacological target is only growing and for which the tools, agonists as antagonists, are always few in number or of partial specificity.
Reviewer 2 Report
The reviewed article is well written and shows interesting results of functional analyses for one of the scorpion venom toxins. However, several questions and concerns arise during the reading process that I would like the authors to address.
The first aspect perhaps stems from the fact that it was necessary to modify the article so that the authors remain anonymous. If so, I hope that the missing elements will appear in the final version of the article. If not, then in my opinion these are deficiencies that should be addressed:
1. Citations 29 and 30 are missing from the text
2. The Materials and methods section in the first paragraph mentions the fraction obtained by ion-exchange chromatography. If this is the result of a previously published experiment, the citation is missing here. If not, a description of the methodology for this step is needed.
3. Information about bioethics committee approvals, approvals for working with animals, etc. is missing.
The second issue is missing results or inadequate description:
1. Peptide masses were determined from MS spectra. Please show the spectra, if only in the supplementary materials
2. In the text one of the peptides is described as 8.6, while in Figure 2 as 8/6# please standardize the nomenclature.
3. The caption of Figure 2 lacks an explanation of the abbreviations of the peptide names to which the sequences were aligned. There is also no description of how this was done in the methods section. Please add information on whether the order of the sequences in Figure 2 matters.
4. In paragraph 2.2, please note the correct notation of genes and proteins. In rats, gene symbols are italicized, first letter upper case and all the rest lower case, while proteins should have the same as the gene symbol, but not italicized and all upper case
5. Paragraph 2.2 also raises my biggest doubt about the proper use of controls. The presence of ANO1 protein was checked by immunolabeling, while the presence of the products of the other genes was checked at the transcript level by qPCR. My biggest doubt is why the expression level of Ano1 was not also checked by qPCR. Although the authors do not directly compare the expression levels between the methods, one cannot clearly say that the expression level of e.g. Ano6 is very low if we do not know at what level the expression of Ano1 is. Perhaps in reality these levels are not that low at all. I think this experiment should be added to this section.
6. Section 2.4 mixes methodology with results. Please move the procedure descriptions to section 4. Please also explain why some experiments used a peptide concentration of 0.1µM and others used 0.5µM?
7. Section 2.5: I am missing information on how it was decided where to introduce mutations in the designed variants of peptide 7.1. Maybe the authors could expand on this in the discussion
8. At the end of section 2.5 the authors mention an experiment using circular dichroism. There is no description of this experiment in the methodology.
The last point is the information that should be corrected/supplemented in the Materials and Methods section.
1. I mentioned earlier the lack of description of ion-exchange chromatography. However, there is also a second inconsistency here. The methodology describes further fractionation of fraction 7, while the results section also mentions fractions 2 and 8. This doubt is also related to the description of the peptide synthesis section. If only peptide 7.2 was analyzed in the long run, why are the other two presented at all, especially since they are omitted from the methodology?
2. Why was detection after capillary electrophoresis done at 200 nm? This is not a standard length for either aromatic amino acids or peptide bonds. Please clarify.
3. There is a typo in the name time-of-flight in line 253
4. Please explain why there is such a large difference in the concentration of carboxypeptidase B used for the different peptides
5. In lines 297 and 339 the citations have the wrong format
6. I believe that the primer sequences used in the qPCR experiment should be added to this section (either as supplementary material)
Author Response
The authors thank the three experts who took the time to provide their expertise to improve the quality of this manuscript. They will find our answers and arguments hereafter in blue and an amended manuscript with the modifications in red.
For all reviewers we want to precise why you have submitted this manuscript. This study began in 1997, the selectivity of toxins from scorpion venom for ion channels, gave us the idea to look for a toxin that could inhibit the activated chloride-calcium conductance of smooth muscle cells. At that time the Ano1 channel had not been identified and it that why we have performed experiment on the cellular model well known in the lab with clearly electrophysiologically identified chloride conductance. This work was presented in several congresses but had not yet been published because we did not find the structural modification of the toxin allowing to derive a radioligand for precisely identifying the target. We have recently reviewed this work and have decided to submit it to your judgment because 1/ the interest of targeting activated chloride-calcium channels of smooth muscle cells is making a comeback (see bibliography of the MS) and 2/ the study by RTqPCR and by immunohistological labeling of the chlorine conductance of the cellular model used suggests that Ano1 is clearly expressed and 3/ our work confirms that the position of this toxin family as a ligand of the channels carrying chlorine conductance or of their direct regulator. Finally, the structural modifications that we have made also suggest a very precise interaction between toxin and target protein showing the subtlety of protein interactions.
Reviewer 2
The reviewed article is well written and shows interesting results of functional analyses for one of the scorpion venom toxins. However, several questions and concerns arise during the reading process that I would like the authors to address.
The first aspect perhaps stems from the fact that it was necessary to modify the article so that the authors remain anonymous. If so, I hope that the missing elements will appear in the final version of the article. If not, then in my opinion these are deficiencies that should be addressed:
- Citations 29 and 30 are missing from the text
Correction done line53.
- The Materials and methods section in the first paragraph mentions the fraction obtained by ion-exchange chromatography. If this is the result of a previously published experiment, the citation is missing here. If not, a description of the methodology for this step is needed.
We have added references.
- Information about bioethics committee approvals, approvals for working with animals, etc. is missing.
We propose to add in the MS a paragraph line 296-300. In France, the euthanasia of animals for cell preparation without any behavioral and surgical protocol does not require specific authorization but of course the animal facility is registrated as indicated in the paragraph : Animals: to isolate the cells, tissues were taken from rats (150 to 250 g of body weight) which were euthanized in a CO2 chamber according to the protocol defined by the manufacturer (CO2 rack, Tem-Sega) in the dedicated room of the animal facility (A32-063-940). All experiments used rats from supplier (Janvier labs) and supernumerary rats produced in the animal facility. Tissues are collected in HBBS medium at 4°C and transported to the laboratory for tissue dissection for cell preparation and RTqPCR experiments.
The second issue is missing results or inadequate description:
- Peptide masses were determined from MS spectra. Please show the spectra, if only in the supplementary materials.
As I add in the introduction of my answers, the experiments were performed in 1997-1998 and now I cannot provide the MS spectra as you required because I have not these rawdata.
- In the text one of the peptides is described as 8.6, while in Figure 2 as 8/6# please standardize the nomenclature.
Figure 2 was modified
- The caption of Figure 2 lacks an explanation of the abbreviations of the peptide names to which the sequences were aligned. There is also no description of how this was done in the methods section. Please add information on whether the order of the sequences in Figure 2 matters.
We add in the legend the source of the different sequences, we have chosen the nomenclature providing from Ali et al., 2016, because we have used this publication to compare the efficient toxin with structurally related toxins
- In paragraph 2.2, please note the correct notation of genes and proteins. In rats, gene symbols are italicized, first letter upper case and all the rest lower case, while proteins should have the same as the gene symbol, but not italicized and all upper case
correction done
- Paragraph 2.2 also raises my biggest doubt about the proper use of controls. The presence of ANO1 protein was checked by immunolabeling, while the presence of the products of the other genes was checked at the transcript level by qPCR. My biggest doubt is why the expression level of Ano1 was not also checked by qPCR. Although the authors do not directly compare the expression levels between the methods, one cannot clearly say that the expression level of e.g. Ano6 is very low if we do not know at what level the expression of Ano1 is. Perhaps in reality these levels are not that low at all. I think this experiment should be added to this section.
The presence of the ANO1 protein was verified by immunostaining, but the presence of other proteins that may have chloride channel functions could only be analyzed by RTqPCR because for some of them the antibodies are not available. We have performed the detection with RTqPCR of Ano1 in portal vein VSMC and add the result expressed as the others in the text line 88. As the reviewer can read now the relative level of expression of tested genes are mentioned in function of their expression level in the text (line 88-92).
- Section 2.4 mixes methodology with results. Please move the procedure descriptions to section 4. Please also explain why some experiments used a peptide concentration of 0.1µM and others used 0.5µM?
We have corrected the question of concentration, in fact it is not clear. I thank you for your comment.
- Section 2.5: I am missing information on how it was decided where to introduce mutations in the designed variants of peptide 7.1. Maybe the authors could expand on this in the discussion
the choice of mutations was guided by the desire to produce derivatives radiolabeled with iodine 125. These are therefore strategic choices making it possible to move the labeling sites without drastically modifying the three-dimensional structural balance of the toxin. We failed to maintain the efficacy of the toxin through these structural changes which led to the cessation of work and led us to propose this work for publication in this state. We have added the explanation in the discussion as you required (line198-200).
- At the end of section 2.5 the authors mention an experiment using circular dichroism. There is no description of this experiment in the methodology.
We add a paragraph about this point (line 283-285)
The last point is the information that should be corrected/supplemented in the Materials and Methods section.
- I mentioned earlier the lack of description of ion-exchange chromatography. However, there is also a second inconsistency here. The methodology describes further fractionation of fraction 7, while the results section also mentions fractions 2 and 8. This doubt is also related to the description of the peptide synthesis section. If only peptide 7.2 was analyzed in the long run, why are the other two presented at all, especially since they are omitted from the methodology?
As mentioned in materials and methods Peptides Lqh 2-2, Lqh 7-1 and Lqh 8-6 were eluted with retention times of 38, 23, 35 min respectively. Lqh7-1 is the only toxin reproduced by synthesis.
- Why was detection after capillary electrophoresis done at 200 nm? This is not a standard length for either aromatic amino acids or peptide bonds. Please clarify.
This is a mistake the detection was performed at 220nm.
- There is a typo in the name time-of-flight in line 253
Correction done
- Please explain why there is such a large difference in the concentration of carboxypeptidase B used for the different peptides.
For Lqh8-6 the protocol has been adapted to obtain a correct sequencing. As you can read the buffer, enzyme concentration and pH and temperature have been adapted.
- In lines 297 and 339 the citations have the wrong format
Correction done
- I believe that the primer sequences used in the qPCR experiment should be added to this section (either as supplementary material)
Primers are commercial ones designed and validated for RTqPCR by Bio-Rad and we used their capacity to manufacture plates. We have indicated the reference of the plate that we have design with their collaboration for this study.
Reviewer 3 Report
The investigators present a study wherein they describe a number of small molecular weight peptides from the scorpion Leuirus quinquestratus hebraeus. Peptides designated as Lqh 2-2, Lqh 7-1 and Lqh 244 8-6 were identified and purified. The peptides were sequenced and then tested in rat portal vein slices. Patch clamp experiments demonstrated that Lqh 7-1 and its synthetic equivalent blocked Cl channels without affecting calcium traffic in the cell.
The number of experiments performed per condition is not provided in Methods, the figure legends, and are scattered in the text of Results – but only incompletely so. The numbers per condition are also random – sometimes n=5, sometimes n=11, etc. There is no statistical justification for this erratic design.
Figure 1 has the x-axis labeled as “Temp”. It should be “Time”.
Figure 5 has red “lighting bolts” pointing at different parts of the various molecules, but the legend does not tell the reader what they represent.
The authors do not provide information concerning how the venom was collected or processed, and the source of the scorpion(s) is also omitted.
The Methods section discusses using the t-test for statistical comparisons, but I cannot find any such comparisons in figures or text.
In summary, this is an interesting work. However, as written, it has materials, experiments and data that cannot be repeated with the information provided. Thus, it is difficult to be able to substantiate the conclusions drawn by the authors with the work presented.
Author Response
The authors thank the three experts who took the time to provide their expertise to improve the quality of this manuscript. They will find our answers and arguments hereafter in blue and an amended manuscript with the modifications in red.
For all reviewers we want to precise why you have submitted this manuscript. This study began in 1997, the selectivity of toxins from scorpion venom for ion channels, gave us the idea to look for a toxin that could inhibit the activated chloride-calcium conductance of smooth muscle cells. At that time the Ano1 channel had not been identified and it that why we have performed experiment on the cellular model well known in the lab with clearly electrophysiologically identified chloride conductance. This work was presented in several congresses but had not yet been published because we did not find the structural modification of the toxin allowing to derive a radioligand for precisely identifying the target. We have recently reviewed this work and have decided to submit it to your judgment because 1/ the interest of targeting activated chloride-calcium channels of smooth muscle cells is making a comeback (see bibliography of the MS) and 2/ the study by RTqPCR and by immunohistological labeling of the chlorine conductance of the cellular model used suggests that Ano1 is clearly expressed and 3/ our work confirms that the position of this toxin family as a ligand of the channels carrying chlorine conductance or of their direct regulator. Finally, the structural modifications that we have made also suggest a very precise interaction between toxin and target protein showing the subtlety of protein interactions.
Reviewer 3 comments
The investigators present a study wherein they describe a number of small molecular weight peptides
from the scorpion Leuirus quinquestratus hebraeus. Peptides designated as Lqh 2-2, Lqh 7-1 and Lqh
244 8-6 were identified and purified. The peptides were sequenced and then tested in rat portal vein
slices. Patch clamp experiments demonstrated that Lqh 7-1 and its synthetic equivalent blocked Cl
channels without affecting calcium traffic in the cell.
The number of experiments performed per condition is not provided in Methods, the figure legends,
and are scattered in the text of Results – but only incompletely so. The numbers per condition are
also random – sometimes n=5, sometimes n=11, etc. There is no statistical justification for this erratic
design.
“n” is the number of cells tested or the number of experiment. We have now precise all the n value in tne text of the results part
Figure 1 has the x-axis labeled as “Temp”. It should be “Time”.
modified
Figure 5 has red “lighting bolts” pointing at different parts of the various molecules, but the legend
does not tell the reader what they represent.
we added in the legend that the red flashes indicate the points of mutations.
The authors do not provide information concerning how the venom was collected or processed, and
the source of the scorpion(s) is also omitted.
Crude venom came from Latoxan, we have added this point in material and methods line235.
The Methods section discusses using the t-test for statistical comparisons, but I cannot find any such
comparisons in figures or text.
Statiscal analyses were precised now in results part.
Round 2
Reviewer 1 Report
The authors have addressed my comments and concerns.
Author Response
thank you for your review. Best regards
Reviewer 2 Report
The authors have addressed my comments and concerns. I believe that in its current form, the maniscript is suitable for publication in Toxins.
Author Response
thank you for your review. Best regards
Reviewer 3 Report
The authors still did not indicate in the new version in the legend of figure 5 what the lightning bolts mean. They mentioned it in their rebuttal, but not in the revision.
Author Response
Please accept my apologies for not having checked the correction of the legend of figure 5. This time it is well done as you can see in red in the new version of the text. Best regards